

# How well does ramped thermal oxidation quantify the age distribution of soil carbon? Assessing thermal stability of physically and chemically fractionated soil organic matter

Shane W. Stoner[1,2], Marion Schrumpf[1], Alison Hoyt[3], Carlos A. Sierra[1,4], Sebastian Doetterl[2],

Valier Galy[5], Susan Trumbore[1]

[1]Biogeochemical Processes Department, Max Planck Institute for Biogeochemistry, Jena, 07745, Germany

[2]Department of Environmental Systems Science, ETH Zürich, Zürich, 8092, Switzerland

[3]Earth System Science, Stanford University, Stanford, 94305, USA

[4]Department of Ecology, Swedish University of Agricultural Sciences, Uppsala, SE-750 07, Sweden

[5]Woods Hole Oceanographic Institution, Woods Hole, 02543, USA

*Correspondence to*: Shane W. Stoner (sstoner@bgc-jena.mpg.de)

## Abstract

Carbon (C) in soils persists on a range of timescales that depend on physical, chemical and biological processes that

interact with soil organic matter (SOM) and affect its rate of decomposition. Together these processes determine the age distribution of C. Most attempts to measure this age distribution have relied on operationally defined fractions using properties like density, aggregate stability, solubility, or chemical reactivity. Recently, thermal fractionation, which relies on the activation energy needed to combust SOM, shows promise in separating young from old C by applying increasing heat to decompose SOM and equate this with biochemical stability in soil. Here, we investigated

radiocarbon (14C) released during thermal fractionation to reconstruct thermal stabilities and age distributions of C released from bulk soil as well as its component density and chemical fractions. We found that bulk soil and all density and chemical fractions released progressively older C as temperatures increased. In addition, all fractions released some C across the entire temperature range, indicating that bulk soil thermal fractionations integrate young and old C at all temperatures. In the bulk soil, age distribution could be identified by isolating particulate C prior to

thermal fractionation of mineral-associated SOM. For the Podzol analyzed here, thermal fractions confirmed that ~95% of the mineral-associated organic matter (MOM) had a relatively narrow 14C distribution, while 5% was very low in 14C and likely reflected C from the < 2mm parent shale material in the soil matrix. By first removing particulate C using density or size separation, thermal fractionation can provide a rapid technique to study the age structure of MOM and how it is influenced by different OM-mineral interactions.

## 1 Introduction

Soil organic matter (SOM) consists of a remarkably complex and diverse collection of organic molecules which exhibit widely varying persistence in soil. Plant tissue chemistry, soil environmental conditions, soil mineral characteristics, physical aggregation, and microbial communities have all been demonstrated to impact the stability

of SOM (Lehmann and Kleber, 2015; Basile-Doelsch et al., 2020; Kleber et al., 2021). These factors collectively influence the age and transit time distributions of the carbon (C) in SOM, making it challenging to link the timescales of OM stabilization and destabilization to the various mechanisms that allow C to persist in soils.

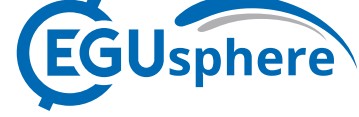

Measurement of soil radiocarbon ([14]C) has been used for decades to describe mean SOM ages. However, the mean [14]C values measured on bulk SOM integrate different pools and stabilization mechanisms and thereby obscure critical information on the distribution of SOM age. By combining timescales from years to millennia, interpretation of bulk [14]C measurements is made more difficult due to integration of [14]C from both natural sources affected by radioactive decay (natural [14]C, integrating multiple centuries to millennia) and [14]C produced by atomic weapons ("bomb" [14]C) that reflect short-term cycling (annual to century) (Trumbore, 2000; Baisden et al., 2013). Disentangling these signals is complex, and requires the integration of [14]C data with models to estimate SOM transit times and ages (Sierra et al., 2018; Metzler et al., 2018).

In an effort to better describe the distribution of age and cycling rates in bulk SOM, a number of physical and chemical fractionation methods have been developed to elucidate how the bulk [14]C can be broken into pools with different amounts of [14]C depending on physical or chemical characteristics (Sollins et al., 2009; Lavallee et al., 2020). In particular, density fractionation, a method that separates SOM associated with denser minerals from low-density 'free' particulate organic matter (FPOM), has demonstrated success in reliably distinguishing faster (low density) from slower (mineral associated) cycling C (Cotrufo et al., 2019; Heckman et al., 2022). However, mineral-associated organic matter (MOM) fractions themselves have been shown by many studies to be comprised of both faster and slower cycling C as evidenced by the change in [14]C content after chemical extraction or oxidation (Jagadamma et al., 2010; Schrumpf et al., 2021) or from tracking bomb [14]C into mineral fractions (examples include Trumbore, 1993; Torn et al., 1997; von Lützow et al., 2007, and more recently Schrumpf et al., 2013; Rasmussen et al., 2018; Heckman et al., 2018). Despite their widespread use and demonstrated utility for separating organic C by age as well as physical and chemical properties, most fractionation methods consume significant laboratory time and resources (Lavallee et al., 2020; Heckman et al., 2022). Further, some treatments remove C in forms such as sodium polytungstate solution that cannot be easily recovered or analyzed for C or [14]C content, meaning that the isotopic signature of removed C must be solved using mass balance constraints.

Ramped pyrolysis/oxidation (RPO), or thermal fractionation, is a relatively new method to functionally fractionate OM in sediments and soils (Rosenheim and Galy, 2012; Plante et al., 2013; Hemingway et al., 2017). This process applies increasing temperature of thermal decomposition as a proxy for the activation energy required to oxidize C, with the assumption that this provides a comparable measure of its resistance to decomposition in the soil environment. The result is a reproducible profile of $CO_2$ released as a function of increasing temperature (thermogram), from which activation energy distributions can be calculated (Hemingway et al., 2017). By collecting the $CO_2$ released over specified intervals as temperatures are continuously increased, "pools" of C with distinct thermal stability can be isolated, collected, and analyzed isotopically (Rosenheim and Galy, 2012). Because all C is released as $CO_2$, it is possible to characterize all of the C in a sample rather than inferring what was removed from analysis of the residual material. A further advantage of such "thermal fractionation" is that it can be compared with pyrolysis-GC/MS of SOM to evaluate how the chemistry of combusted SOM also changes with activation energy. Previous studies have shown that the breakdown of lipids and polysaccharides releases C at lower temperatures, while thermal decomposition of phenolic and aromatic compounds dominate at higher temperatures (Quénéa et al., 2006; Grandy et al., 2009; Sanderman and Stuart Grandy, 2020). Thus thermal





fractionation has the potential to define the [14]C (age) distribution of organic C and relate that to the activation energy and chemistry of the OM in a soil sample.

Several studies have investigated soils using oxidative thermal fractionation (Plante et al., 2013; Grant et al., 2019; Hemingway et al., 2019). Compared to sediments, where these methods have been more widely applied, soil thermograms release a greater proportion of the total C over a narrower temperature range and have lower variation in age across thermal fractions (Hemingway et al., 2019). This may reflect a broader set of OM sources

in sediments that can include eroded soil containing very old and highly processed C as well as fresh material from aquatic organisms.

Typically, C released from both sediments and soils by thermal oxidation also increases in age with temperature of combustion, i.e. activation energy, confirming linkages between SOM bonding characteristics and the mechanisms

of C stabilization (Plante et al., 2011; González-Pérez et al., 2012). However, different SOM stabilization mechanisms or local environments can complicate the interpretation of activation energy - age relationships; for example the same chemical compound sorbed to different mineral substrates can have very different activation energies (Feng and Simpson, 2008). Thermal oxidation of OM not associated with minerals, such as dissolved organic C (DOC), oxidizes at narrow but relatively high temperature ranges, possibly contributing young C at high

temperatures that would be mixed with C released from mineral fractions at the same temperature (Grant et al., 2019; Hemingway et al., 2019). Given the wide range of [14]C ages measured in various physical and chemical fractions, and the potential for recycling of C in soils through microbial processing, we expect some range of ages of C within each bulk soil thermal fraction.

Here, we apply oxidative thermal fractionation to SOM previously separated using physical (density) and chemical (extraction and oxidation) methods. Using mass balance approaches, we describe the contribution of each fraction to bulk soil thermograms and [14]C signatures. We also present the first thermal fractionation results using a commercially available instrument for conducting thermal fractionation. Our goals were to determine (1) the degree to which the physically and chemically separated fractions represent mixtures of OM with different

activation energies and [14]C distributions; (2) to determine the [14]C distribution of C contained in physically or chemically separated fractions; (3) to assess the viability of thermal fractionation as an alternative to more time intensive lab methods in determining the [14]C distribution of SOM.

## 2 Methods

### 2.1 Site description and density fractionation

Soil material used in this study was sampled from a Podzol developed on granitic parent material under spruce forest in central Germany (Schrumpf et al., 2013, 2021). This soil was selected because it was already known to have large differences in [14]C content between density fractions (Schrumpf et al., 2021) and because of strong depth-dependent stabilization processes in Podzol A and B horizons (De Coninck, 1980). Surface (0-10 cm) and

subsoil (30-50 cm) samples were subjected to laboratory fractionations described in detail by Schrumpf et al. (2013). Briefly, soils first underwent density separation using dense sodium polytungstate solution (SPT) (1.6 g/cm[3]). Suspended OM was separated from denser material that did not float using centrifugation. The floating



free particulate OM (FPOM) fraction was collected and rinsed to remove remaining SPT solution. The sinking fraction was dispersed again in 1.6 g cm$^{-3}$ solution and sonicated to disrupt aggregates, then centrifuged. After centrifugation, floating material from the supernatant was collected, rinsed and designated as occluded particulate organic matter (OPOM). The remaining dense material in the sediment was repeatedly rinsed to remove SPT solution and is designated mineral associated organic matter (MOM).

### 2.2 Chemical fractionation

Two chemical fractionations were performed in parallel on the MOM fraction, as described by Schrumpf et al. (2021). The first subsample was extracted with NaF-NaOH to solubilize and remove all potentially desorbable SOM complexed with minerals through pH increase and competition with OH- and F- anions (Kaiser et al., 2007; Mikutta and Kaiser, 2011). Briefly, 125 mL of a NaF-NaOH solution was added to 25g MOM material, agitated overnight, and centrifuged. The supernatant was extracted, and an additional 125 mL of NaF-NaOH was added to repeat this process four times in total. Then, each extraction was filtered through glass fiber filters and combined. The remaining soil material was washed with deionized water and freeze-dried.

The second chemically treated MOM underwent strong oxidation in heated hydrogen peroxide ($H_2O_2$) to isolate the most resistant and oldest OM (Helfrich et al., 2007; Jagadamma et al., 2010). In this procedure, 60 mL $H_2O_2$ was added to a mixture of 2 g MOM and 20 mL deionized water. Samples were then heated and periodically stirred in a 50˚C water bath for a total of 120 hours. Samples were then centrifuged, washed with deionized water, freeze dried, and homogenized with a ceramic ball mill.

### 2.3 Thermal fractionation and method development

Oxidative thermal fractionation of bulk SOM and physically and chemically separated fractions was performed using an Elementar soliTOC Cube carbon analyzer. Samples were not analyzed under pyrolytic conditions, as pyrolysis can produce charring artefacts, and $^{14}$C distributions have been shown to be comparable between operational modes (Williams et al., 2014; Grant et al., 2019). The design of this instrument is very similar to those used in previous thermal fractionation publications (Rosenheim and Galy, 2012; Bianchi et al., 2015). Primarily, it consists of two ovens in sequence, a mechanical arm to hold and manipulate the sample container, and an non-dispersive infrared analyzer (NDIR) to measure the $CO_2$ concentration in the gas exiting the ovens. The sample is introduced to the first oven, which is heated at a constant rate under a constant flow of carrier gas supplied through the sampler arm (78% $N_2$, 22% $O_2$). The second oven contains a Pt catalyst held at 800˚C that ensures all C released from the sample is oxidized to $CO_2$. The carrier gas then passes through a glass tube filled with brass wire at 20˚C to remove HCl from acidified samples (no samples were acidified in this experiment) followed by a glass tube containing magnesium perchlorate to remove water vapor. Finally, $CO_2$ concentration in the gas mixture is measured by the NDIR (DIN 19539).

Several additional procedures were required to adapt use of the commercial device for collection of C released by thermal fractionation. Due to the relatively large sample size (> 1g of dried soil or fraction) required to collect small thermal fractions with sufficient C for radiocarbon measurement, and the high flow rate of carrier gas in this instrument, samples with high C content (such as standards or FPOM/OPOM fractions) were diluted to ~2% (by



weight) C with pre-combusted sand (1000˚C for 10 hours) to prevent ignition during heating. An artificial soil standard was analyzed with different sand dilutions to ensure that thermograms were not altered by dilution with sand (Supplementary fig 3). Further, the sample oven was designed for rapid heating (up to 110˚C min$^{-1}$), and temperatures were observed to be less stable at slower heating rates. To reduce the cycles of on/off oven cycling while ensuring thermogram consistency (with sand dilution), samples were heated at 15˚C per minute.

Thermograms and activation energy were analyzed using the open-source "rampedpyrox" Python package (Hemingway 2016; Hemingway et al. 2017). For each thermogram, a distributed activation energy model derived from time-temperature C-release data is solved inversely to produce a continuous distribution of activation energy ($E$, in kJ mol$^{-1}$). It assumes a finite set of $n$ components (thermal fractions, in order of increasing temperature, referred to as F1 - F$max$, where F$max$ is thermal F$n$, the highest temperature range collected (Appendix tables 1 and 2)) in superposition to construct the bulk soil $E$ distribution. Each of these components can thus be mathematically assigned a mean activation energy ($\mu E$) and standard deviation ($\sigma E$). Here, standard deviation describes the variance of distribution of $E$, or the heterogeneity of the bonding environment, within a thermal fraction or sample, rather than data variance. Thus, direct comparisons can be made between $E$ distribution within a thermal fraction and its isotopic composition. However, it should be noted that such activation energy descriptors derived from thermograms are not necessarily comparable to other methods of measuring activation energy (Feng and Simpson, 2008; Hemingway et al., 2019).

To collect $CO_2$ for isotope analysis, a custom collection manifold was attached to the instrument outflow port (Supplementary figs. 4-6). The manifold consists of parallel glass $CO_2$ traps submerged in LN$_2$ under vacuum. Exhaust gas released within a desired temperature range (thermal fraction) flows through a cold trap until the desired upper temperature is reached. Then, the trap is closed and the next opened to collect the next $CO_2$ fraction. This process is repeated for each thermal fraction (F1 (first thermal fraction) – F$max$ (highest temperature thermal fraction), see Appendix tables 1 and 2). A vacuum pump together with a capillary restriction upstream of the manifold was used to reduce the overall pressure in the manifold system to < 6 mbar to improve cryotrapping efficiency and to prevent condensation of $O_2$ in the LN$_2$ traps.

Traps with $CO_2$ samples were subsequently transferred to a vacuum line where the $CO_2$ was further purified (see below) and measured volumetrically for comparison (calibration) of the NDIR $CO_2$ analysis. An aliquot was taken for analysis of $\delta^{13}C$ using a modified gasbench inlet to a continuous flow IRMS (Wendeberg et al., 2013). In addition to $CO_2$, we noticed that nitrogen oxide gasses (including $N_2O_3$, which is dark blue when frozen) were visibly trapped. These gasses are produced by the reaction of $N_2$ and $O_2$ at high temperatures. As these, as well as S oxides that also freeze with $CO_2$ at liquid nitrogen temperatures, can cause graphitization failure, we used an additional purification procedure to remove them. An amount of sample $CO_2$ representing approximately 0.5 mg C was transferred cryogenically and then sealed under vacuum in a pre-combusted pyrex tube containing ~50 mg CuO and ~10 mg Ag (Hemingway et al., 2017) and baked at 525˚C for one hour. Purified $CO_2$ released after breaking this tube was graphitized using zinc reduction (Xu et al., 2007) and measured at the Keck AMS lab at University of California Irvine. Radiocarbon data are expressed as Fraction Modern (Fm).



## 3 Results

To our knowledge, this was the first thermal fractionation procedure performed using a commercial C analyzer.
We thus first present results on the performance and reliability of this setup to demonstrate the viability of this
method to future researchers (Sections 3.1-3.3). We then describe data on SOM decomposition as a function of
temperature, modeled activation energy, and isotopic signatures of thermal fractions within and between density
and chemical fractions (Sections 3.4-3.7) and compare this to thermal fractionation of the bulk soil.

### 3.1 Method testing and quality assurance

### 3.1.1 Reproducibility of the thermograms

An artificial soil standard containing calcium carbonate was repeatedly analyzed ($n = 6$) to determine consistency
and reproducibility of themograms on commercially available equipment (Supplementary fig. 1). The bulk soil and
fractions analyzed experimentally here released >99% of C below 600˚C. In the critical $CO_2$ collection range
between 100 and 600˚C the average standard deviation of C released for a given temperature range was +/- 2.2%
of the mean C released within that range (Supplementary fig. 1). The standard deviation between repeated standard
soil samples over the entire temperature range, including the calcium carbonate peak between 650 and 800°C,
averaged +/- 2.9%.

Another test of error was to compare the bulk soil thermograms with the summed thermograms of component
density fractions (see Figure 1b, below). The general agreement of bulk and summed thermograms suggest that
there is no significant alteration of thermal stability during fractionation and that density fractions may be
compared to bulk soil.

### 3.1.2 Accuracy of radiocarbon analyses

We analyzed [14]C standards with known isotopic composition to assess the degree to which extraneous C was
added in our combustion and trapping procedures that could change the isotope signatures of analyzed samples. To
assess how much extraneous C with low amounts of [14]C ('dead' C) was added, we analyzed a standard with [14]C
values containing mostly 'bomb' C (Chinese Sugar Char, diluted with precombusted sand, UCI Consensus
measurement Fm 1.353 +/- 0.003, $n = 55$) and achieved final values of 1.355 +/- 0.009 ($n$=3). Not included in this
average are many analyses made while refining the overall method that tended to be lower (up to Fm 0.034 below
accepted values). However, in the configuration used for the soil analyses presented here, values were within Fm
0.007 of the known values. To assess whether extraneous modern C was added, we analyzed coal with zero [14]C,
diluted with pre-combusted sand. The Fm averaged 0.006 +/- 0.001. The amount of 'extraneous' C was also
assessed by analyzing only pre-combusted sand that should contain no C, and measuring the amount of $CO_2$
trapped after the full combustion procedure. Across the whole temperature range, this measured 0.026 mg C with
average Fm 0.9766 ($n$=6), representing in most cases 0.5% (for 5 mg total C collected) of the total combusted
sample. Such "blank" values were applied for correcting [14]C values reported here, and the blank C and [14]C was
distributed across all thermal fractions proportionally based on temperature range.

### 3.1.3 Mass balance of thermal fractions



Finally, our confidence that our method produces reliable and repeatable measurements of C content and isotopic composition was evaluated through successful mass and isotope balance. The amount and isotopic signatures of C estimated by summing the various fractions compared well with the bulk soil measurements (Figure 1b, Appendix tables 1 and 2). For example, summing C-weighted Fm $^{14}$C from the three density fractions (FPOM, OPOM, MOM) for the 30-50 cm depth interval yielded 'bulk' Fm of 0.815, slightly below the measured bulk soil value of 0.824. Replicate analysis of bulk soil from 30-50 cm yielded Fm values of 0.819 and 0.815, and 0.812 from seven thermal fraction measurements including high temperature tail fractions.

**Table 1:** Summary information of bulk soil and fraction thermal stability and isotopic compositions, including activation energy (E) indices.

| Depth | Fraction | Fraction Percent of Total C | $E$ (kj mol$^{-1}$) | $\sigma E$ (kj mol$^{-1}$) | Whole Frac. Fm | Max Thermal Fm[a] | Min Thermal Fm[a] |
|---|---|---|---|---|---|---|---|
| 0-10 cm | Bulk Soil | - | 134.1 | 14.2 | 0.997 | 1.048 | 0.751 |
| 0-10 cm | FPOM | 8.7 | 133.5 | 15.3 | 1.080 | 1.102 | 1.067 |
| 0-10 cm | OPOM | 6.2 | 135.3 | 14.0 | 0.992 | 1.040 | 0.968 |
| 0-10 cm | MOM | 85.1 | 133.7 | 15.8 | 0.985 | 1.037 | 0.728 |
| 0-10 cm | NaF Res. | 28.8 | 137.8 | 18.2 | 0.912 | 0.959 | 0.761 |
| 0-10 cm | H$_2$O$_2$ Res. | 13.5 | 136.3 | 12.8 | 0.859 | 0.868 | 0.781 |
| 30-50 cm | Bulk Soil | - | 138.7 | 14.0 | 0.824 | 0.854 | 0.323 |
| 30-50 cm | FPOM | 15.6 | 141.8 | 15.9 | 1.087 | 1.085 | 1.064 |
| 30-50 cm | OPOM | 8.2 | 144.3 | 14.7 | 0.847 | 0.869 | 0.822 |
| 30-50 cm | MOM | 76.3 | 137.9 | 16.1 | 0.786 | 0.791 | 0.230 [b] |
| 30-50 cm | NaF Res. | 29.9 | 137.9 | 24.7 | 0.713 | 0.798 | 0.562 |
| 30-50 cm | H$_2$O$_2$ Res. | 15.5 | 141.2 | 17.7 | 0.628 | 0.753 | 0.414 |

[a]: Maximum and minimum $^{14}$C content collected via thermal fractionation within the sample

[b]: Value calculated by mass balance, +/- 0.02 Fm

### 3.2 Thermal analysis of physically and chemically fractionated organic matter



Having demonstrated the accuracy and reproducibility of our thermal analysis system, we next compared the thermograms and the isotopic ($^{14}$C and $^{13}$C) signatures of $CO_2$ released as a function of temperature for each physical and chemical fraction individually, then compared the summed contribution of each physical/chemical fraction to the bulk soil (for density fractions) or MOM (for chemical fractions) to assess (1) the behavior of each of the different fractions and (2) how much each fraction contributes to the bulk thermogram at different temperature intervals.

### 3.2.1 Thermograms and activation energy

All density and chemical fractions and bulk soil released 90-98% of their total C between 150 and 500˚C. No fraction had a unique thermal signature (Figures 1a, 1b), and the thermograms mostly overlapped, with some C released across the whole temperature range of combustion. However, differences were observed among density fraction thermograms. For particulate fractions (FPOM and OPOM), C release displayed one or two muted peaks and most of the C was oxidized between 250 and 450˚C. MOM and chemical residues released most of their C between 250 and 350°C, but also released more C at temperatures >500°C compared to FPOM and OPOM fractions. Since most bulk soil C is in the MOM fraction (Table 1), thermograms for the bulk soil resemble those of the MOM fractions in both depths (Fig. 1a).

Mean activation energy ($\mu E$) estimated from thermograms of bulk soil and fractions ranged from 133.5 to 137.8 kJ mol$^{-1}$ in surface soil and 137.9 to 144.3 kJ mol$^{-1}$ in subsoil (Table 1, Appendix tables 1 & 2). Between depths, $\mu E$ was greater in subsoil than surface soil by ~5-9 kJ mol$^{-1}$ for all samples except NaF extraction residues, which showed no difference. In subsoil, particulate fractions FPOM and OPOM $\mu E$ values were significantly greater than bulk soil and MOM. Standard deviation of $E$ ($\sigma E$), a metric of bond strength heterogeneity, only varied significantly ($p < 0.05$) with depth among chemical fraction residues, which suggest greater diversity of bonds in the subsoil (Hemingway et al. 2017). Thus, despite large differences in the chemistry and relationship to mineral surfaces, the activation energy range was similar across all chemical and physical fractions. It is puzzling that NaF and $H_2O_2$ residues had lower activation energies than might be expected, given that they represent the most "recalcitrant" C resistant to harsh chemical treatments.







Figure 1: Relative magnitudes of thermograms, as C released as a function of temperature, with fractions scaled to their relative contribution to the total C in each panel. A: Bulk soil and density fraction thermograms, for 0-10 cm and 30-50 cm, respectively. Density fraction (FPOM, OPOM, MOM)

thermograms are scaled to their relative contribution to bulk soil C (Table 1). Dashed lines represent summed thermograms of the three density fractions. Comparison of summed and bulk thermograms show good agreement and suggest that fractionation procedures do not significantly alter the thermal stability of component fractions. B: Thermograms of MOM and chemical fractionation residues. The difference between MOM and given chemical fraction thermograms represent the thermal profile of C removed by the

chemical treatment (NaF-NaOH or $H_2O_2$). Chemical fraction residue thermograms are scaled to their relative residual C content of the MOM fraction. C: Proportional contribution of density fractions to bulk soil C released across collection temperature range (colored fill). Solid black line represents bulk soil thermogram to highlight total C release from bulk soil at each temperature. Density fractions are cut off when C release is no longer discernible from instrument IR-detector background noise.



**Figure 2**: **Thermograms with radiocarbon measurements. Top) 0-10 cm, Bottom) 30-50 cm. Left-hand column Y-axis values represent contribution to the total (bulk soil) C. NaF Res. and H2O2 Res. panels are scaled in proportion to their total C contribution to MOM. Color scale indicates the Fraction Modern (Fm) of the C released in each temperature range; the scale is doubled above Fm 1 to emphasize difference between post-bomb C (Fm >1.0) and C that has had significant radioactive decay (Fm <1).**





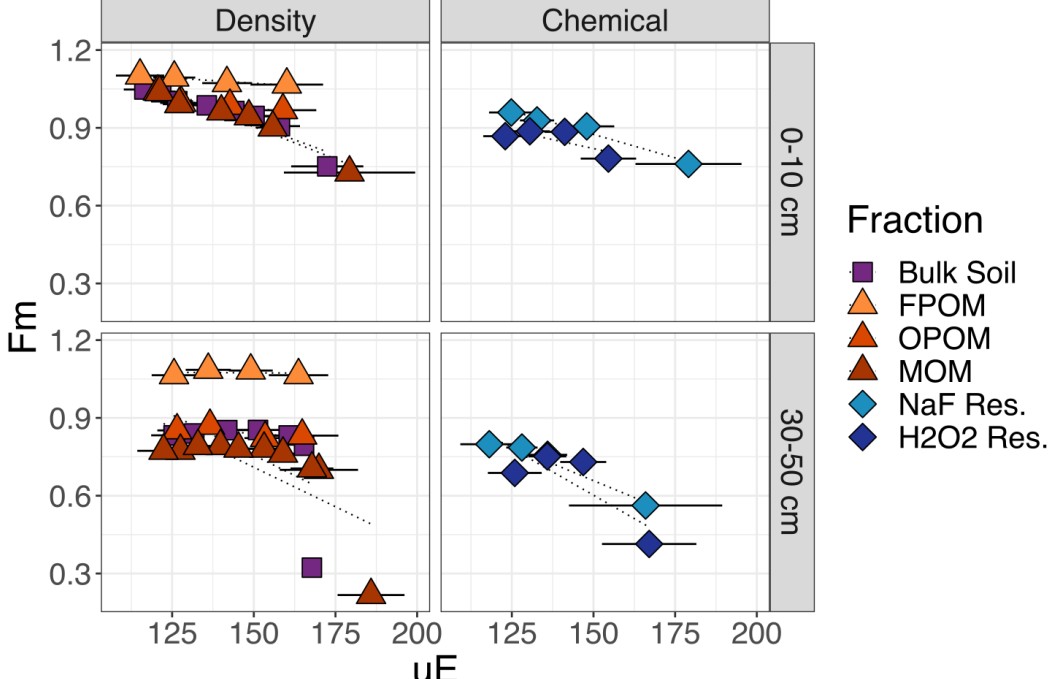

**Figure 3**: **Radiocarbon (Fm) as a function of activation energy for C collected across different temperature intervals from combustion of bulk soil, compared with those of combusted component density and chemical fractions. Horizontal bars represent $\sigma E$ for each thermal fraction, which indicates the range of activation energies represented by a given thermal fraction  Right-hand labels denote depth in cm.**





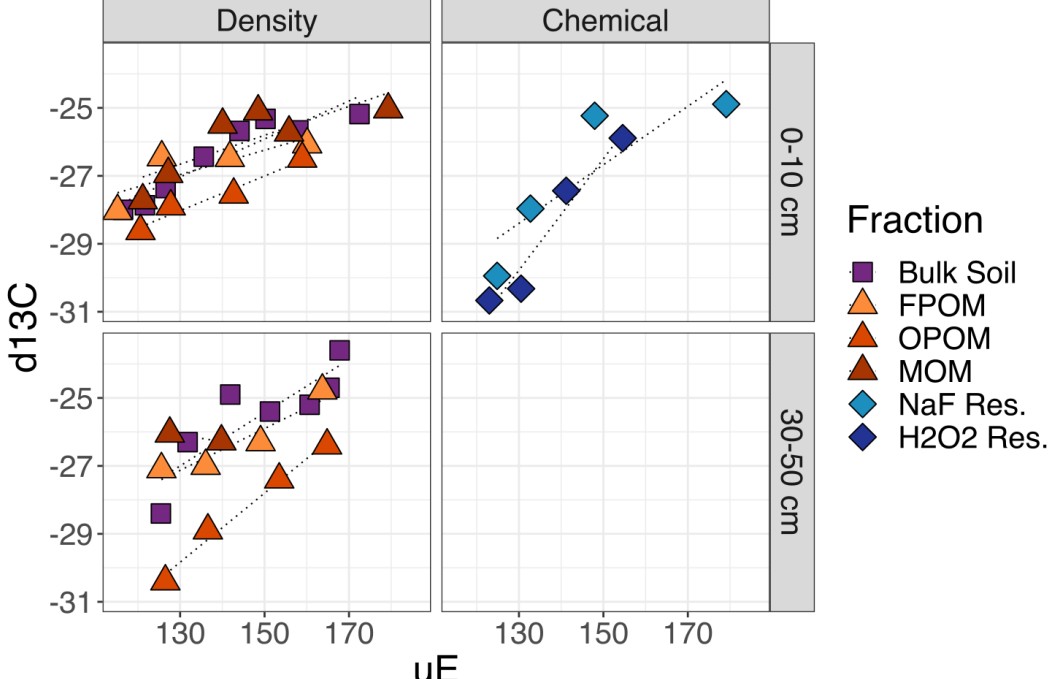

**Figure 4**: **δ¹³C measured for each fraction as in Figure 3. Low C content and limited sample material prevented data collection from some fractions (MOM, NaF Res., H₂O₂ Res. in subsoil). Right-hand labels denote depth in cm.**

### 3.3 Radiocarbon

The mean radiocarbon ($^{14}C$, expressed as Fm) differed for each density or chemical residue fraction (Table 1). For a given soil depth, the FPOM had the highest $^{14}C$ content, consisting mostly of C fixed since the 1960's (Fm >1.0), while the lowest $^{14}C$ was in the residue after $H_2O_2$ treatment of the MOM. The $^{14}C$ of the bulk soil and each

fraction decreased from the 0-10 cm to 30-50 cm depth, and the overall pattern of Fm for the different physical and chemical fractions (FPOM>OPOM> MOM> NaF residue > $H_2O_2$ residue) remained the same.

Within all fractions, the Fm of released $CO_2$ stayed similar or declined as the temperature increased (Figure 2; temperatures of combustion are converted to activation energy in Figure 3). In both Figures 2 and 3, it is clear that the large differences in $^{14}C$ between the FPOM other density and chemical fractions are larger than the range of

Fm released across temperatures during combustion of the individual fractions. Indeed, as reported by Schrumpf et al. (2021), much of the combusted C from MOM fractions has similar $^{14}C$ signatures (small range of Fm), except for the highest temperate/*Ea* fractions of MOM and Bulk soil.

For the bulk soil and MOM fraction in the surface sample, and FPOM fractions at both depths, the C oxidized at the lowest temperature had Fm >1, indicating that a portion of the C in the fraction was fixed mostly in the last 60

330    years. For the FPOM fractions with Fm >1, $^{14}C$ values are not as simply related to 'age' of the C. For example, the most recently fixed C could have lower values than the mean, but so could older C if that is a mixture of pre-and





post-bomb C. For all samples other than FPOM, the decline in Fm [14]C indicates a clear trend of increasing age (decreasing Fm, indicating more time for radioactive decay of [14]C) especially at temperatures above that where most C was released (Figure 2). The highest-temperature thermal fractions (F*max*, mostly 450 - 800°C, Appendix tables 1 and 2) of surface bulk soil and MOM were similarly depleted in [14]C and significantly older than any other values measured (Figs. 2 & 3, Appendix Table 1) .

In subsoils (30-50 cm), bomb [14]C was found only in the FPOM fractions, so the decline in [14]C with energy was determined mostly by the much lower [14]C in C released at high temperatures (Figure 3). All fractions except NaF Res. increased in Fm from the C collected in F1 and F2 (and F3 in bulk soil) temperature ranges (140-375°C), followed by decreases at increasingly higher temperatures. Excluding FPOM and OPOM, all fractions decreased significantly in Fm in F*max* compared to the temperature range previous.

The chemical fractionation residues contained C with lower Fm than the unextracted MOM at all temperature ranges except in the highest temperature range collected. However, the highest temperature fraction collected for the MOM was greater (505 - 750˚C), because insufficient C evolved from the chemical fraction residues in this range (Figure 3). Thermograms for the chemical residues follow a similar pattern to those of MOM, with a small amount of younger but chemically resistant C released at low temperatures, and much older C released in F*max*. As noted above, although the chemical residues contained less than 30% of the total MOM C (Table 1), their thermograms were very similar. The very old F*max* thermal fractions in the chemical residues represent only a small amount (1-4%) of the total bulk soil C (Appendix tables 1 & 2).

### 3.4 δ[13]C

The δ[13]C of $CO_2$ released from SOM generally increased with temperature in bulk soil and all fractions. The range of δ[13]C values from F1 to F*max* was the greatest (increasing by 4-5‰) for the chemical residues, and smaller (3-4‰) for the density fractions. Across density fractions, the range of values and the differences in δ[13]C between different fractions was greater in the deeper soils. Interestingly, the FPOM at 30-50 cm was more enriched in [13]C than OPOM. Subsoil δ[13]C was generally more enriched at high temperatures than surface soil.

### 3.5 Contributions of different physical fractions to the thermal oxidation of bulk SOM

Thermograms (Figure 1) demonstrate that C released by the bulk sample at all temperatures contains C contributed from all physical and chemical fractions. For example, of the bulk C released in the temperature range where most C was released  (250 to 325˚C), FPOM and OPOM contributed 9% and 6%, respectively, of total C released in surface soil, and 16% and 8% in subsoil (Table 1, Fig. 1a). However at higher temperature ranges, while the total C released was small (<5% of the total C) the proportional contribution from FPOM and OPOM fractions increased to ~40% in surface and 30% in subsoil (Figure 1c).

Thus, each thermal fraction from a combusted bulk soil contains C with a broad range of Fm and [13]C, with variable contributions from the different physically fractionated components. Figure 5 summarizes the Fm distribution of C across the density and thermal fractions, and emphasizes that the difference of Fm between density fractions (especially FPOM versus MOM) is greater than the range of Fm within any individual density



fraction (excluding a small amount of very old MOM) released as a function of temperature (activation energy)
       (Figure 5).

       The measured distribution of $^{14}$C for C released with increasing temperature from the bulk soil clearly does not
       capture the contribution of FPOM with high Fm, because its young C is released across the same temperature

ranges as other density and chemical fractions (Figs. 1 a,c, 3, 4). Thus, the surface soil age distribution misses the
       ~9% of total C in FPOM that has a much higher $^{14}$C signature than bulk soil; instead its contributions skew the
       bulk soil thermal $^{14}$C (Figure 5, wide bars in the middle of the distribution) higher than the separated MOM
       thermal fractions (green). This difference is even more pronounced in the subsoil.

With a sufficient number of thermal fractions at high temperatures, bulk soil C captured the small percentage of C
       with very depleted $^{14}$C signatures better than the chemical fractions that still mixed younger and older constituents.
       In surface soil, bulk soil F$max$ $^{14}$C values (Fm 0.75) were comparable to F$max$ fractions of NaF Res. and H$_2$O$_2$
       Res. (Fm 0.76 and 0.78, respectively), and represented similar amounts of C (2.6%, 2.7%, and 3.0% of total C,
       respectively) (Appendix table 1). Bulk subsoil F$max$ isolated older C (Fm 0.32, 5% of total C) than F$max$ values of

NaF and H$_2$O$_2$ residues (Fm 0.56, 8.1% total C and Fm 0.41, 3.8% total C, respectively), but high-temperature
       samples were not collected for these fractions because of low C yield (Appendix table 2).

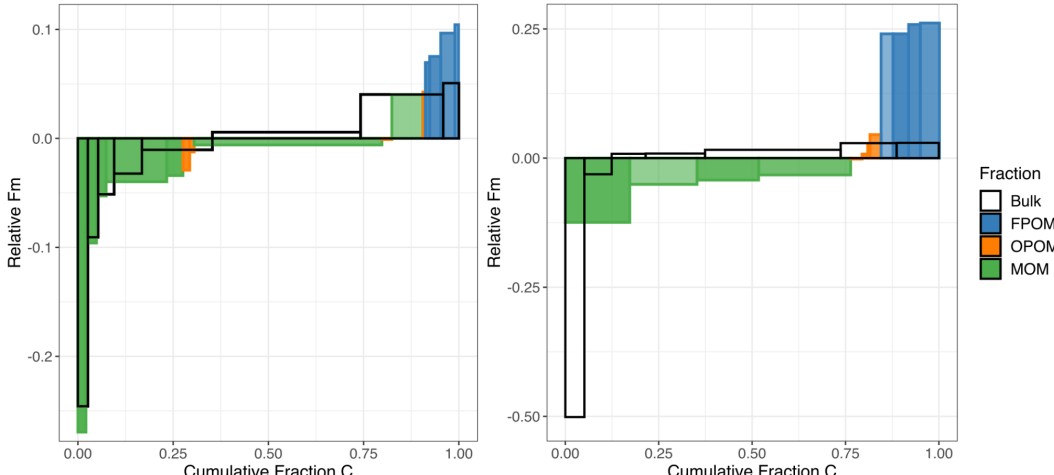

**Figure 5: Comparison of the cumulative Fm distribution of C released during thermal fractionation of bulk
soil versus oxidation of physically and chemically separated density fractions in the topsoil (left; 0-10cm)**
**and subsoil (right; 30-50cm). The height of each histogram element represents the Fm $^{14}$C, normalized to
       the overall bulk Fm value. Effectively, values above 0 contain more $^{14}$C than bulk soil, and values below 0
       contain less. The width of bars corresponds to the proportion of total soil C in the fraction. The unfilled
       histogram elements (no color) represent thermal fractions from the bulk soil, while the colored bars
       represent the thermally fractionated FPOM, OPOM and MOM fractions shown in previous figures. Darker**
**colors within a fraction correspond to higher temperature/$E$ fractions, and lighter colors reflect
       cooler/lower $E$ fractions. Both are ordered by the $^{14}$C content, with lowest on the left and highest on the
       right.**




## 4 Discussion

In this work, we compare the thermal oxidation profiles and [14]C distributions of thermally fractionated SOM with more frequently applied methods using physical (density) and chemical separation methods in a Podzol at two different depths. Thermal fractionations of bulk soils and component physically and chemically separated SOM fractions demonstrate that increased thermal stability (i.e. higher activation energy) is associated with lower radiocarbon ([14]C) content, i.e. older C ages, and more enriched [13]C content. We find that when measured individually, all density and chemical fractions released C across most of the OM oxidation temperature range

(100-600°C), indicating that each fraction contains a spectrum of activation energies and isotopic signatures. Thus bulk soil thermal fractions mix the contributions from C pools contained within each of the density and chemical fractions. Because there are large differences in Fm between the physically and chemically separated fractions, this means that the C released with similar activation energies in thermal analysis mixes C with different chemistry and age. Together, these results provide insights into the utility of standard and thermal fractionation methods for

capturing SOM age structure.

### 4.1 Physically and chemically separated fractions are mixtures of organic matter with different activation energies and [14]C distribution

#### 4.1.1 Activation energy can predict age within a fraction but not between fractions

Each physically and chemically separated fraction released older, more [14]C depleted, and more [13]C enriched C with increasing temperature and activation energy (Figs. 4, 5). This supports the general assumptions and results of thermal analysis: that older and more microbially processed/degraded C will be released at increasing temperatures, even among fractions like FPOM that are not associated with minerals (Plante et al., 2009).

Particulate fractions FPOM and OPOM release C across a broad range of temperatures, reflecting energies required to oxidize the molecules that make up plant material (e.g. cellulose) that release C between ~300 and 500°C (Dahiya and Rana, 2004; Plante et al., 2009). Both FPOM and OPOM fractions are likely to contain fresh plant material and microbial residues (Angst et al., 2021; Castanha et al., 2008). However, despite a range of activation energies, $\delta^{13}C$ signatures (Fig. 4), and high $\sigma E$ (Table 1) all suggesting chemical diversity, FPOM in this

soil is all of recent origin (post-bomb, Fm >1.0) (Fig. 3) and breaks down within decades. Mineral associated (MOM) fractions demonstrated larger though mostly overlapping ranges of activation energy, releasing very old and [13]C enriched C at temperatures greater than 550°C (Figs. 2-4). Thus, while within a given fraction there are predictable patterns of increasing age and $\delta^{13}C$ with activation energy, these patterns do not allow prediction of age from activation energy alone. This highlights fundamental differences between activation energy, [14]C

signature and age. While activation energy can either increase or decrease over time as C moves between different molecular forms with the progress of decomposition and recycling, the age of the involved C atoms can only increase over time. For most MOM fractions, with Fm <1.0, a decline in Fm reflects the loss of [14]C due to radioactive decay and therefore indicates an increase in age. However, for the FPOM fractions, and increase or decrease in Fm is more difficult to associate directly to a specific age due to the temporal dynamics of the bomb

spike. Thus, it is difficult, and perhaps not to be expected, to associate activation energy directly to [14]C.



**4.2 Age distribution of C from thermal oxidation of each physically and chemically isolated fraction is less than the differences between fractions**

**4.2.1 Chemical and thermal methods both show that MOM in this soil reflects two mechanisms operating on different timescales**

Comparisons of the chemical and thermal fractionation methods for MOM both indicate the presence of two distinct components with very different Fm, one representing >95% of the C and having Fm similar to that of the bulk MOM, a small amount <5% of much older C. In this Podzol, the main stabilization mechanisms are likely the

445 interactions between percolating dissolved organic matter and pedogenic (oxy)hydroxides that could explain the large amount of relatively younger C (decades to centuries) that represents the largest thermal fraction of the MOM (F2) that is also removed by NaF and $H_2O_2$ (Figure 1b). As shown by Schrumpf et al. (2021), the chemical extraction and oxidation of MOM using NaF and $H_2O_2$, respectively, removed C that was slightly higher in [14]C concentration than the MOM overall, leaving a small but much older residue that resists destabilization. The $H_2O_2$

oxidation removed more C and left an older residue than the NaF extraction (Table 1). However, Fm [14]C of the extracted C was the same for the two procedures (Schrumpf et al., 2021). Thus the majority of removed MOM-C had similar, younger ages that could reflect SOM more weakly associated with mineral surfaces, while the small proportion remaining could have been trapped within the mineral structure (e.g. in clays on formation) or represent elemental C. Both methods support the idea put forward by Schrumpf et al. (2021) that much of the MOM was

cycling on century timescales while a small amount (<10%) was much older (Fm 0.628). However, thermal methods demonstrate that the 3% of subsoil MOM oxidized at temperatures greater than 505°C was even older (Fm 0.23, Fig. 4)

While NaF and $H_2O_2$ treatments removed younger C, the combustion of the residues showed that they still

contained C with a range of activation energies and ages. These results are somewhat puzzling, as particularly the $H_2O_2$ treatment is expected to remove all but the hardest to oxidize (i.e. "recalcitrant") C. The chemical methods used here are believed to only remove sorbed C that has higher Fm (i.e. is younger) than the residue (Kaiser et al. 2007; Mikutta et al. 2010). The observation of diverse activation energies in residue C could indicate incorporation of sedimentary parent material C into microbial food webs (Seifert et al. 2013). We therefore expected that the

$H_2O_2$ residue would not only be older, but also on average have higher activation energy. On the contrary, there was actually less C in F*max* for both residues compared to the unextracted MOM (Figure 1, 3), such that the oldest C in the residues was likely mixed with younger C.

**4.2.2 Potential sources of oldest C**

Understanding the nature of the small amount of very old C found in MOM and bulk soil, and explaining the age and $\delta^{13}C$ structure of the NaF and $H_2O_2$ residue thermal fractions, requires additional information. One possibility is that the oldest C is derived from the shale parent material of the Wetzstein site (Schrumpf et al., 2011). Unpublished [14]C data collected from the surface of rock fragments found in the soil indicate a Fm of 0.27, similar to values calculated for subsoil MOM F*max* fractions (Table 1). The thermal alteration of sedimentary parent

material during metamorphosis could also explain the chemical recalcitrance, heavier $\delta^{13}C$, and higher activation energies of this very old C.



A second possibility is the presence of non-crystalline minerals that are often correlated with the amount of very old C found in soil (Huang et al., 2016; Khomo et al., 2017; Heckman et al., 2018). Wetzstein soils have moderate oxalate extractable Fe contents of 9.2 (0-10 cm) and 17.4 (30-50 cm) g kg$^{-1}$ (Schrumpf et al. 2021 Biogeosciences, supplement). Dithionite extractable Fe concentrations (including both crystalline and non-crystalline components) were 17 and 27.4 g kg$^{-1}$ (respectively). However, quantifying such effects would require investigation of soils with varying amounts of non–crystalline minerals.

A third explanation of long SOM persistence is the stochastic nature of the decomposition process. Available C is not uniformly decomposed, and some substrate may persist in soil on much longer timescales (Sierra et al., 2018). Through random chance associated with biological, chemical, and physical processes, a small portion of total SOM remains in soil for centuries to millennia. Such obviously persistent C may be associated with the high activation energies measured here.

### 4.3 How well do thermal fractions describe age distributions of SOM compared to standard laboratory methods?

#### 4.3.1 Thermal fractionation cannot separate youngest C

As noted above, physical and chemical fractionation methods are successful in separating C into fractions with very different mean ages. However, thermal fractionation of these SOM fractions revealed age structure within each one. Standard fractionation procedures require large quantities of expensive SPT solutions for density separation, as well as substantial effort and time required in the laboratory (Cotrufo et al., 2019). Mass balance is often difficult due to dissolution of C or minerals in the SPT and chemical solvents. In contrast, thermal fractionation has advantages that include capturing all sample C in the thermal fractions and requiring minimal sample preparation. One of our goals was to assess how well thermal fractions could be used to determine the age distribution of SOM. Therefore we made a detailed comparison of the C age distributions using thermal fractions of the isolated fractions to thermal fractionation of the bulk soil (Figure 5).

The goal of any fractionation scheme is to provide a better idea of the age distribution of C in soils that integrate a number of different kinds of SOM and stabilization mechanisms. Such age distributions can be used to constrain models of SOM dynamics (Sierra et al., 2014), and test hypotheses linking stabilization mechanisms with rates of C cycling.

Overall, Figure 5 demonstrates that thermal fractionation of bulk SOM fails to capture the youngest part of the age distribution. This is because the youngest component of the soil C, the low density FPOM, releases C across nearly the entire range of combustion temperatures, making the C released at the lowest temperature too old, and the C released at higher temperatures too young. At the highest temperatures, however, thermal oxidation methods can isolate C even older than what can be found using harsh chemical extractions (Fig 3). At the very highest temperatures, the contributions of C from oxidation of FPOM and OPOM are relatively small (Figs. 1c, 2) but may skew data with much younger C. In order to best capture the age distribution of C in SOM, we therefore





recommend first separating the low density fraction (perhaps also separable by size as well as density, if the main point is to remove relatively fresh plant material) (Castanha et al., 2008; Lavallee et al., 2020), then applying thermal fractionation with attention to C liberated at very high temperature and using this to determine the age structure of MOM.

### 4.4 Future Applications

Describing the distribution of ages in SOM is a powerful tool for testing hypotheses about the timescales of different C stabilization mechanisms in soils, and for comparison with age distributions produced by multi-compartment models (Metzler et al., 2018; Chanca et al., 2022). Our results are for a single soil, a Podzol that likely has one major mechanism for stabilizing C on mineral surfaces: interaction with pedogenic oxides. To explore other mineral stabilization mechanisms and timescales, it would be useful to compare thermograms and age distributions for soils with different mineral composition - e.g. allophane, 2:1 clays, 1:1 clays, sands, and then with mixed mineralogy soils. Additionally, comparison with temperature-resolved spectra (e.g py-GC/MS, (Sanderman and Stuart Grandy, 2020), DRIFTS (Nkwain et al., 2018), etc.) that associate SOM chemistry with thermal stability may help to determine the roles that OM chemistry and mineralogy play in controlling C age and persistence in soil.

### 5 Conclusions

Commercial instruments can be used for thermally fractionating SOM with consistent repeatability. Advantages of thermal methods include greater throughput, low sample volume, and the ability to characterize all of the C in a sample. Our method was shown to reproduce lab standards and standard soil thermograms and isotopes, and allowed mass and isotopic balancing of soil thermal fractions.

Each density and chemical fraction contained OM with a spectrum of ages. The MOM fraction displayed two distinct age components in this podzol, identified in both soil depths: likely the younger component that represents the majority of MOM stabilized by association with pedogenic (oxy)hydroxides, and the older component possibly inherited from shale parent material.

We conclude that thermal fractionation cannot completely replace standard fractionation methods to connect SOM properties (e.g. activation energy) to age distributions. Fresh FPOM contributes C of fairly consistent age across temperatures up to 550°C and thus dilutes the signal of older C. This method was effective at identifying multiple stabilization timescales on the MOM fraction, which contains the majority of C cycling on relevant time scales in most soils. We thus recommend separating and measuring [14]C of FPOM, then analyzing thermal fractions of MOM to help distinguish faster- and slower-cycling mineral associated components. Further efforts are needed to explore thermograms and [14]C distributions of MOM fractions with very different mineral stabilization mechanisms.

### Acknowledgments

We gratefully acknowledge our funding sources and the Biogeochemistry journal reviewers. We thank Dr. Axel Steinhof and Dr. Xiaomei Xu for their invaluable assistance in developing the equipment and methods used in this



study. We also thank the staffs of the Max Planck Insitute for Biogeochemistry, UC Irvine, and the Woods Hole Oceanographic Insitute for their assistance in radiocarbon sampling and data preparation. Finally, we thank the CarboEurope Project for access to the archived soils used in this study.

**Author Contributions**

SWS and ST designed, constructed, and tested method hardware and protocols. Data were collected by SWS and analyzed by SWS and MS with input from all authors. SWS lead the writing of the manuscript with significant contribution from ST and input from all authors.



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
