# Peer review of "How well does ramped thermal oxidation quantify the age distribution of soil"

_EGUsphere, 2022_

## Author Response (AR1)

**Author Response #1**

This study examines mechanisms of soil organic matter (SOM) persistence. The manuscript is well written, the experimental design is sound (though limited to a single soil), and the conclusions drawn are consistent with the data collection. The study is particularly compelling because it pulls together two older or more conventional methods for studying SOM persistence (fractionation and radiocarbon) with newer energetic approaches using ramped combustion thermal analysis. The authors have demonstrated where the data from these various sources clearly overlap and can be interpreted consistently, and more importantly, where the data appear to be inconsistent or in conflict. The conclusion that I reached in reading the manuscript is that physical and chemical fractionations to not isolate pools of SOM that are consistent with our conceptual representation of them. I'm not sure the authors would agree, but if so, this point might be made more clearly and succinctly. Overall, I think the manuscript is worthy of publication, but have several specific comments and suggestions for revision outlined below.

We thank Dr. Plante for his kind comments and suggestions for improving this manuscript. The data we present do indeed present challenges to standing interpretations of density fractions, especially in the "mineral-associated" (MOM) phase. This is not the first article to hypothesize that SOM is dynamic and variable. For example, while the "slow" pool, so to speak, most likely exists in the MOM fraction, it would be misleading to think that the MOM fraction *is* wholly the slow pool. This was displayed here by comparing soil horizons. In addition, we see similar $^{14}$C values in the highest temperature fractions of FPOM as the lowest-temperature fractions of MOM. There exists evidence for both fast and slow cycling C associated with minerals, and we agree that the call to action for re-conceptualizing SOM, and MOM in particular, should be stronger in this manuscript. We have placed a greater focus on age distributions as a better metric for C dynamics than mean C ages, and this method's capacity to determine age structure.

Abstract

Ln22-25: These two sentences and the following one seem to be in the wrong order. The authors might want to rethink the logical sequence of ideas they are trying to convey.

We understand the logic of this comment. We rearranged the sentences to flow a bit more logically, moving the first sentence between the second and third, and shifting a clause. For explanation, the original order reflects the order of the discussion, where it is introduced and described in a more logical fashion.

Ln24: Would "is better" a more precise wording than "could be" for what the authors are trying to assert? They seem reluctant to be prescriptive, but I think that reluctance reduces the impact of the study.

We appreciate this comment and the confidence it conveys. "Is better" is certainly true to describe the result, and has been added. "Could be" was written in the past tense, as in "was", but is indeed more passive.

Introduction

Ln32: Adverbs like "remarkably" are too subjective and don't do any narrative work. It can be removed or elaborated to describe what is remarkable if the authors feel it is important. This is true throughout the manuscript.

Thank you for this comment. It has been removed (as the sentence already has a more useful adverb), and such words have been selectively removed throughout the text.

Ln33: Can the author be more precise than "varying" persistence? For instance, might a range of ages/MRTs be provided to demonstrate this variance?

We have added "(hours to millennia)", and a citation to the book "Radiocarbon and Climate Change" edited by Schuur et al., 2016 which has a nice table on page 169.

Ln39: I think this gets adequately elaborated later in the paragraph, and I'm also not well versed in the nuances of 14C analyses, but because soils are open systems I thought we were supposed to avoid terms like "age" and instead invoke residence/transit time. Perhaps provide more precise diction here?

Here we use the term " mean age", in the sense suggested by Sierra et al.. 2018, and used by others as a simple way to allow readers to get an idea of the average time elapsed since C in a given fraction was fixed from the atmosphere.  As written in the

text, this is based on a one pool model where all C has equal probability of decomposition and using it to estimate the mean decomposition rate. This also provides readers with a way to determine how the differences in 14C with thermal fractions translate into differences in mean C turnover.

In the past, it has been recommended to avoid the term 'residence time', because in open systems such as soils one must distinguish between the age of mass inside the system and the age of mass leaving the system. The terms of system age and transit time capture these two concepts, respectively, and this is elaborated better in further paragraphs. To avoid jargon in the first paragraphs, we simply refer to the age of carbon in the soil and the age of carbon in the microbial respiration flux.

L35-36: "These factors collectively influence the age distribution of carbon (C) in SOM and the age of C in the microbial respiration, making it challenging to link the timescales of OM stabilization and destabilization to the various mechanisms that allow C to persist in soils."

Throughout the manuscript, the authors suffer a degree of "recent-ism" when it comes to citations about physical and chemical fractionation. While the work by Cotrufo and colleagues is excellent, some might not consider it seminal because it reframes and revisits much of the work that was done ~10 years prior. For instance, Gregorich et al. (2006) is among a number of excellent papers that clearly conceptualizes and defines "uncomplexed" organic matter and elaborates on physical fractionation. Similarly, there is excellent work published by EA Paul and SE Trumbore on 14C work on chemical fractionation residues that is uncited in favor of more recent work. While citation of recent work is not a problem in itself, it can be problematic when presented as seminal when it may not be.

Thank you for the suggestions, we have included them and others throughout the manuscript.

Methods

Ln160: I believe figures are to be numbered in order of mention in the text. The author might consider referring to Figs S1 and S2 here, or renumbering. Also see my comments below about Results section 3.1.

Thank you for pointing this out. They have been re-ordered in the supplement and re-named in the text.

Ln164-175: I recommend moving this paragraph to ln196-197. This paragraph is about data and it interrupts the narrative about the instrumental and analytical methodology.

The narrative might instead be about the collection of observational data, followed by the manipulation and analysis of this data.

Thank you for point this out as well. We agree that the paragraph flows better at the end of the section, since it is more about interpretation than lab method.

Ln166: Values for the fitting parameter lamba are important in determining the shape of the energy distribution curve. These values should perhaps be reported for reproducibility.

This is a good point! They have been added to the supplementary materials, Table S1.

Ln167: The notation around "E" can be a bit confusing. There is E, µE, σE and Ea. Be sure that the nomenclature is well defined at the outset and consistent through the manuscript.

We combed through and removed an "E", and added the "a" as a subscript, from Ea to $E_a$ as is the generally used notation. Furthermore, $E_a$, $\mu E$, and $\sigma E$ were italicized throughout.

Results

Ln199: Unfortunately, this assertion is untrue. I quick search of "soliTOC" will reveal a few recent papers using the instrument for thermal fractionation of SOM. Perhaps the authors need to be more precise if they mean the coupling with 14C analyses.

Section 3.1: While the QA/QC of the method development is very important, I would strongly recommend moving it to the supplemental materials. Reproducibility and accuracy is essential and no other part of the study would be valid without it, but I find that it doesn't adequately contribute to the core narrative developed in the discussion and conclusion to warrant leaving it in the main body.

This manuscript is indeed not the first to present an SOM thermal analysis study on this equipment, but it is the first (to our expanded knowledge) that uses it for thermal fractionation / RPO / "dirt burning" to collect evolved gas for [14]C analysis. We are comfortable with the request (from both reviewers) to move the method QAQC to the supplement, but have chosen to leave some extra detail in the main text.

Table 1: Units in the headers should be "J", not "j". Also, should "E" not be "µE"?

Thank you, this has been corrected.

Ln259 (and 415 and 494): It's not clear to me why third level numbering is needed if there is only one item in the list. Perhaps the structure and numbering scheme can be revisited.

Thank you for the suggestion. It's true, the first paragraph doesn't stand alone and justify the sub-sub-header. The first paragraph was slightly reduced, the 3.2 header expanded to "Thermograms and activation energy of physically and chemically fractionated organic matter", and the 3.2.1 header removed completely.

Ln274: The authors invoke a p-value without any prior description of the approaches/methods for statistical analyses.

We have modified the text to reflect the actual relevant statistical analysis. Because there are two depths with paired samples, we applied a paired $t$-test to compare μE and σE across depths. Due to the minor role that such statistics played in the overall analysis and interpretation, we included the type of test applied in-line with parentheses instead of adding such a short paragraph to the methods section.

"Between depths, μE was greater in subsoil than surface soil on average by 5.2 kJ mol-1 ($p = 0.01$, paired t-test) for all samples except NaF extraction residues, which showed no difference..."

Figure 1: Axis title for the dependent axis is missing. While it is clear from the caption what it should be, the title should be in the figure. Also, the unit for the independent axis should be "°C", not "C".

Thank you very much for this comment. We have updated numerous aspects of Fig. 1, including using density for the first two rows of figures, as this better reflects the distribution of data over temperature. The degree symbol has also been added to the x axis.

Figures 3 and 4: Perhaps the axis titles could be elaborated/defined for clarity? Eg, "[14]C Fraction Modern (Fm)" and "Activation Energy (μE)". Also, replace the letters with greek letters/symbols in the axis titles.

Thank you for the suggestions. The y axis has been updated in this way. The x axis has been rewritten as "$\mu E$ (kJ mol$^{-1}$)" which hopefully solves the issue brough up in this comment.

Ln327: I believe the authors mean "temperature" instead of "temperate".

Thank you for pointing that out. It has been corrected.

Discussion

Overall, I found the discussion to be compelling but a little overelaborated or structured. The complex observations and data do result in a complex set of interpretations, but I think the reader would benefit from a shorter, clearer and more succinct discussion. Here is the message I got when reading the discussion as a possible guide for improving clarity. Placed in our current conceptual framework, physical and chemical fractionations generate pools of SOM with predictable/anticipated mean ages, but not anticipated distributions of ages or "persistence" as expressed by thermally derived activation energies. This calls into question either fractionation or thermal analysis as a means of assessing SOM stabilization mechanisms and persistence. Observations seem to suggest that density or chemical dissolution/oxidation don't generate fractions consistent with our predictions based on conceptual framework. This calls to mind Smith et al (2002) on measured fraction vs. model pool. It seems fractionation may not successfully separate SOM into the distinct pools we hope for. In the end, thermal fractionation can successfully elucidate age distributions of SOM, but works best when free POM is removed.

We agree with the assessment of the discussion laid out here. Thermal fractionation is a useful tool in conjunction with other fractionation methods, but it cannot fully replace them (at least not in separating FPOM adequately). We are happy to trim and focus the discussion as suggested. There are areas that may get further into details than is strictly required for the story conveyed by these data.

Ln505: I'm not sure I agree. Rather than integrating stabilization mechanisms, some fractionation methods are designed to isolate pools of SOM based on specific mechanisms (eg, mineral-associated vs. uncomplexed; easily oxidized vs. recalcitrant). Perhaps this paragraph needs to be rephrased or rethought.

We believe that the phrasing led to a bit of confusion. We intended the meaning of the sentence to be that the total distribution of ages integrates and is controlled by many processes, and fractionations (for [14]C and other analyses) are designed to isolate distinct pools with operational distinctions that can provide information about physical and chemical stabilization mechanisms.

Conclusions

I guess the sense that the authors are restraining themselves from being too prescriptive and are writing prudently. I would encourage providing the reader with a clearer (if not stronger) message based on the outcomes of the study. Concisely but

specifically, where do the authors think the problems/shortcomings/fallacies lie, and what can be done to avoid them?

We thank the reviewer for this comment and call to action. We added some more descriptive language to the existing text (e.g., "This method was effective at identifying multiple stabilization timescales on the MOM fraction, suggesting complex dynamics that may react across multiple timescales including those relevant to climate and management change.")  as well as a sentence that we feel puts a stronger edge on the same message, mainly: "This additional fractionation helps to go beyond using a mean $^{14}$C value towards characterizing $^{14}$C distributions that can provide a more comprehensive description of SOM cycling and potentially a more stringent test for models."

**Author Response #2**

The authors studied samples from a Podzol using ramped thermal oxidation in conjunction with analysis of the 14C signature of the released CO2 and calculation of the corresponding activation energies; all for bulk soil and physical and chemical soil fractions. This is a unique and novel approach as combining 14CO2 with thermal analysis has only recently advanced.

The overall study design and analytical quality of the work is very good; data and results are presented in a clear and concise manner. Most of what I'd comment on has been addressed already by the first reviewer. In particular, I agree with the suggestion of moving parts of the quality control of the experiment to a supplement. Overall I recommend the manuscript for publication in BG, after consideration of a few additional points:

We are very grateful for the positive comments and appreciate the time and effort the reviewer contributed to the improvement of this manuscript, and are happy to hear that it was largely in shape for publication. We agree that the quality control can be moved to a supplement, and have done so. We have also changed some sentences to be more strongly stated in an attempt to be more prescriptive.

We have already made edits in response to Referee #1, and responded specifically to Reviewer 2's comments below

Line 89. This sentence is not clear to me. What is exactly meant by 'linkages between SOM bonding characteristics and the mechanism of C stabilization'?

We thank the reviewer for pointing this out, and upon revisiting the comment agree that it could be said more specifically. "Persistence" is a better term than "bonding characteristics". We have changed the sentence to the following:

"Typically, C released from both sediments and soils by thermal oxidation also increases in age with temperature of combustion, i.e., $E_a$, confirming linkages between SOM persistence and the mechanisms of C stabilization (Plante et al., 2011; González-Pérez et al., 2012)."

Line 157. C-rich fractions were diluted with sand to avoid ignition during heating. This is a bit confusing – isn't the goal of such a dilution to prevent charring during heating?

This is indeed correct. While practically the dilution was done to avoid ignition and loss of samples, it also (and equally importantly) helped to prevent charring artefacts. The text has been updated to mention charring in equal measure (l. 171)

Line 425 ff. The authors emphasise that the pattern of increasing age with activation energy does not allow prediction of age from Ea alone, which highlights fundamental differences between Ea, 14C signature, and age. Although this finding is new in the context of this type of study, it is in a way to be expected and also trivial. Any thermogram of recent (in terms of age) and moderate complex organic material such as wood or grass shows a wide range of activation energies, related to the molecular structure of the material.

Thank you for this comment. It reflects one of the difficult aspects of utilizing radiocarbon. Here, we meant to convey that similar activation energies produce very different ages across fractions for exactly this reason. To make this connection more obvious, we have edited the text as follows. We hope it strengthens and clarifies the point.

L 550-553: "However, as found in other studies (Leifeld and von Lützow, 2014; Williams et al., 2018; Hemingway et al., 2019), these patterns do not allow prediction of age from Ea alone, highlighting fundamental differences in the processes controlling Ea, 14C signature, and age in each fraction."

Interpretation of old 14C signatures at higher temperatures of reaction. Authors refer to a possible contribution of shale, but likewise old charcoal, which is typically thermally (and biologically) quite stable, could contribute and may be mentioned.

Thank you for making this point. It is especially true in forested systems such as this, and we have added brief mention of charcoal at line 610, with two citations to Cusack et al., 2012 and Sanderman et al., 2016. These describe some phenomena of charcoal preservation that are not "inert", but do add some old C. However, the very great age of this C leads us to believe that the shale is still the dominant contributor.

Table 1. Activation is in kJ per mole; Fm should be explained in the table heading.

Thank you for pointing these out. The activation energies have all be corrected to kJ mol$^{-1}$, and an explanation for Fm added to the Table 1 caption.

Figs. 3 and 4. Unit X-axis is missing.

Thank you for pointing this out as well. Figures 3 and 4 have been redone to properly utilize the Greek symbols and add units.